# Impact of Effective Roughness Length on Mesoscale Meteorological Simulations over Heterogeneous Land Surfaces in Taiwan

**Fang-Yi Cheng [1],\*, Chin-Fang Lin [1], Yu-Tzu Wang [1], Jeng-Lin Tsai [2], Ben-Jei Tsuang [2] and Ching-Ho Lin [3]**

[1] Department of Atmospheric Sciences, National Central University, Taoyuan 32023, Taiwan; littlefrogbb@gmail.com (C.-F.L.); sylvia8404@gmail.com (Y.-T.W.)

[2] Department of Environmental Engineering, National Chung-Hsing University, Taichung 40227, Taiwan; jltsai0408@gmail.com (J.-L.T.); tsuang@nchu.edu.tw (B.-J.T.)

[3] Department of Environmental Engineering and Science, Fooyin University, Kaohsiung 83102, Taiwan; PL018@fy.edu.tw

\* Correspondence: bonniecheng18@gmail.com; Tel.: +886-3-4227151-65508

**Abstract:** The Weather Research and Forecasting (WRF) modeling system obtains the aerodynamic roughness length ($z_0$) from a land use (LU) lookup table. The effective aerodynamic roughness length ($z_{0eff}$) was estimated for the island of Taiwan by considering the individual roughness lengths ($z_{0i}$) of the underlying LU types within a modeling grid box. Two $z_{0eff}$ datasets were prepared: one using the $z_{0i}$ from the default LU lookup table and the other using the observed $z_{0i}$ for three LU types (urban, dry cropland and pasture, and irrigated cropland and pasture). The spatial variability of the $z_{0eff}$ distribution was higher than that of the LU table-based $z_0$ distribution. Three WRF sensitivity experiments were performed: (1) dominant LU table-based $z_0$ (namely, S1), (2) $z_{0eff}$ estimated from the default $z_{0i}$ (namely, S2), and (3) $z_{0eff}$ estimated from the observed $z_{0i}$ (namely, S3). Comparisons of the thermal field, temperature, and surface sensible and latent heat fluxes revealed no significant differences among the three simulations. The wind field overestimation and surface momentum flux underestimation in S1 were reduced in S2 and S3, and these improvements were more prominent over areas with highly heterogeneous land surface conditions.

**Keywords:** surface heterogeneity; effective roughness length; land use; surface fluxes; WRF

## 1. Introduction

In mesoscale meteorological modeling, the accuracy of the land surface parameters is important for adequately simulating the flux exchange processes between land and air [1]. For example, Lam et al. [2] studied the effects of six land surface parameters, namely, the roughness length, thermal inertia, soil moisture availability, albedo, surface heat capacity, and surface emissivity, on the meteorological simulation of Hong Kong; the results suggested that soil moisture availability is the most important parameter controlling the flow patterns and surface fluxes. In addition, Cheng and Byun [3] updated land use (LU) data using NASA's LANDSAT-derived satellite products for the Houston Ship Channel area and showed improvements in the boundary layer mixing processes and local wind patterns.

The aerodynamic roughness length ($z_0$) is one of the most important land surface parameters for simulating the wind profile and estimating the momentum, heat, and moisture fluxes in the atmospheric surface layer. Studies have noted that it is necessary to consider the dimensions and spatial distributions of the roughness elements over a heterogeneous surface to estimate $z_0$. Lu et al. [4] calculated $z_0$ over several land surfaces using three years of experimental data from the Xiaotangshan Experiment

(Changping District of Beijing, China). The individual $z_0$ values over each surface component (patch) were first calculated with the characteristic parameters of the roughness elements (vegetation height, leaf area index, etc.), and then the $z_0$ value for the whole experimental field was aggregated using the footprint weighting method. Consequently, $z_0$ was found to vary with the wind direction over a heterogeneous surface.

To capture fine-scale meteorological processes, the parameterization of land surface processes and the consideration of surface heterogeneity are required [5]. Approaches, such as the urban canopy model (UCM), consider subgrid-scale inhomogeneous surface fluxes using a "tile" approach over urban areas [6,7]. Lee et al. [8] applied the Weather Research and Forecasting (WRF)-Noah model in Houston and the surrounding areas and compared the simulation results with and without the application of the UCM; the simulation without the UCM significantly overpredicted the near-surface air temperature (TA) and boundary layer development over the urban areas of Houston. This overestimation occurred because land surface parameters are dependent on the dominant LU type, and for dominantly urban cells, this dependency significantly suppresses the latent heat flux (LHF) and enhances the sensible heat flux (SHF) and storage heat flux. With the use of the UCM, the subgrid approach, which considers the existence of urban vegetation, successfully reduced the warm temperature bias in the urban areas of Houston. In another study, Li et al. [9] developed a mosaic/tiling approach in the WRF version 3.5 coupled with Noah modeling framework by considering different tiles within each grid cell based on land cover categories. A comparison with observed data showed a better performance using the mosaic approach; in addition, the mosaic approach generated simulation results that were more consistent among the different resolutions than did the dominant LU approach.

The consideration of subgrid-scale variability in $z_0$ is an important issue for numerical weather prediction. Taking the island of Taiwan as an example, its Central Mountain Range (CMR) runs from north to south across the entire island. Throughout the CMR, the major land cover type is forest. Figure 1 shows the topographical height distribution and the locations of Central Weather Bureau (CWB) surface weather stations. An accurate representation of $z_0$ over such complex terrain needs to account for the subgrid variations in the underlying roughness elements. Hence, the effective roughness length ($z_{0eff}$), a weighted average of the individual roughness lengths that compose the heterogeneous terrain, is often used [10].

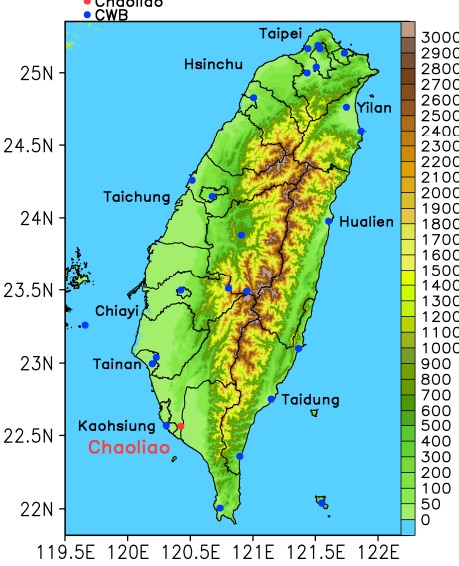

**Figure 1.** Shaded elevation map where the colors represent the terrain height (unit: m). The names of major metropolitan cities are identified. The blue circles mark the locations of Central Weather Bureau (CWB) surface weather stations. The red circle marks the location of Chaoliao.

In this study, we targeted the role that $z_{0\text{eff}}$ plays in mesoscale WRF meteorological modeling and focused particularly on its impact on heterogeneous land surfaces in Taiwan. The $z_{0\text{eff}}$ values for the Taiwan area were determined by considering the individual roughness lengths ($z_{0i}$) of the underlying LU types that compose the modeling grid box. Hence, the values of $z_{0i}$ are dependent on the corresponding LU types. The LU data utilized in this study were reclassified from 2009 Moderate Resolution Imaging Spectroradiometer (MODIS) satellite data products with a spatial grid spacing of 500 m. Detailed descriptions of the reclassified MODIS LU data and the estimated $z_{0\text{eff}}$ values are provided in Section 2.

Several WRF sensitivities were assessed using the lookup table-based $z_0$ and updated $z_{0\text{eff}}$. The aims of this study were as follows: (1) to compare the impacts of the LU table-based $z_0$ and the estimated $z_{0\text{eff}}$, which considers surface heterogeneity, on mesoscale meteorological simulations; (2) to understand the role of the newly derived $z_{0\text{eff}}$ in the meteorological simulation of Taiwan, particularly over areas where the land surface is highly heterogeneous; and (3) to improve predictions of the TA, wind speed and surface turbulent fluxes through the better characterization of surface roughness elements. The simulation results from this study will be used for air quality modeling assessments, which require accurate meteorological modeling results.

Descriptions of the simulation episode, observations and model configuration are described in Section 3. A discussion of the simulation results and a comparison with measurements are provided in Section 4. Finally, some conclusions are given in Section 5.

## 2. Descriptions of the Land Surface Datasets

### 2.1. LU Data

The U.S. Geological Survey (USGS) 25-category LU dataset available in the WRF model is not used in this study because the data are outdated (being from the year 1990); in particular, some of the components originate from a dataset compiled in the 1970s. Moreover, these data do not reflect the expansion of urbanization in Taiwan over the past several decades, and the delineation of the CMR in this dataset is inaccurate [11]. To remedy this problem, new LU data were reclassified using the 2009 MODIS satellite products following the method of Cheng et al. [11]. The MODIS LU data have a grid spacing of 500 m. These data were subsequently aggregated to a model grid spacing; Figure 2 shows a comparison among the LU types at a grid spacing of 3 km. In the USGS dataset, the majority of the LU types in Taiwan are classified as irrigated cropland or mixed forest; consequently, irrigated cropland covers almost two-thirds of Taiwan and is distributed mostly along the western plains, even across the CMR. Nevertheless, such a distribution is not realistic and does not conform to the actual LU distribution. Therefore, the specification of $z_0$ from the USGS lookup table would underestimate the surface heterogeneity in Taiwan and misrepresent the surface roughness elements therein. In comparison with the USGS data, the MODIS LU data accurately depict the locations of the major metropolitan areas, namely, Taipei, Taichung, Kaohsiung, Yilan, Hualien, and Taidong (refer to Figure 1 for these city locations), as well as the distributions of irrigated cropland and forested areas. Furthermore, the LU types in the CMR are accurately identified as forestland and are closely related to the topographic variation. Accordingly, in this study, MODIS LU data are used to provide the underlying land surface conditions and roughness element information.

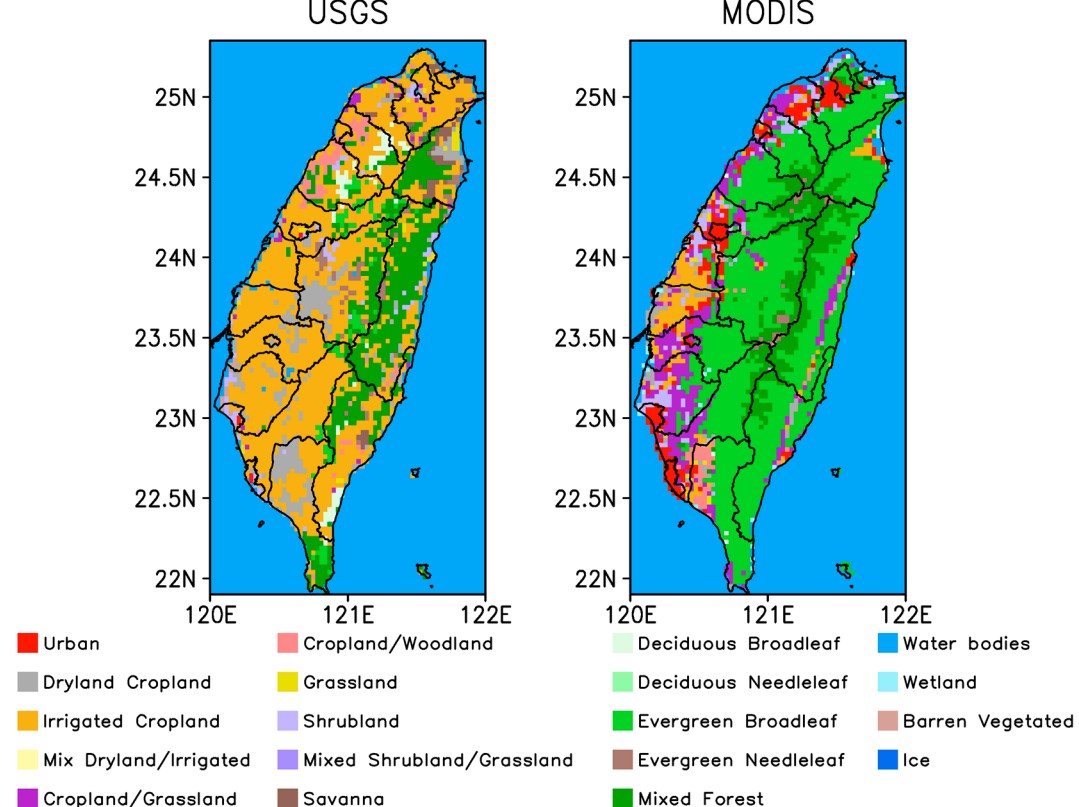

**Figure 2.** Comparison between the land use (LU) type distributions: default U.S. Geological Survey (USGS) 25-category distribution (**left**) and reclassified distribution from the 2009 Moderate Resolution Imaging Spectroradiometer (MODIS) satellite data (**right**).

## 2.2. Aerodynamic Roughness Length

The $z_{0\text{eff}}$ values were derived based on the research performed by Tsai and Tsuang [12] and Tsai et al. [13] following the method of Goode and Belcher [14]. In Goode and Belcher [14], $z_{0\text{eff}}$ was estimated by considering the individual local roughness, $z_{0i}$, of the underlying LU types that compose the modeling grid box; moreover, the concept of a blending layer was introduced to obtain the spatially averaged surface stress. Across the blending layer, the stress varies smoothly from a local surface value to a value aloft, which is adjusted to the patchy terrain [14]. Consequently, the $z_{0\text{eff}}$ values over a sequence of $N$ types of roughness elements can be calculated according to Equation (1) [14].

$$1 = \sum_{i=1}^{N} r_i \frac{\left[\ln\left(\frac{h}{z_{0\text{eff}}}\right) + \frac{h_{ei}}{h_i - h_{ei}} \ln\left(\frac{h_i}{h_{ei}}\right) - 1\right]^2}{\left[\ln\left(\frac{h_i}{z_{0i}}\right) + \frac{h_{ei}}{h_i - h_{ei}} \ln\left(\frac{h_i}{h_{ei}}\right) - 1\right]^2} \tag{1}$$

Here, $z_{0\text{eff}}$ can be determined based upon $z_{0i}$, the internal boundary layer ($h_i$), and the equilibrium layer height ($h_{ei}$) in the grid box. Moreover, $h_i$ is the layer that develops downstream due to changes in the surface roughness, and $h_{ei}$ is the layer in which the flow is effectively adjusted to the local roughness and in which the local stress is constant with height and equals the surface value. The numerical simulations performed by Goode [15] suggest that the depth of $h_i$ after a smooth-to-rough change in roughness is only slightly greater than that after a rough-to-smooth change in roughness. These simulations demonstrate that the depth of the internal boundary layer is determined mainly by a relatively large roughness length and is less sensitive to a relatively small roughness length. Goode [15]

then shows that scaling of the momentum equation yields the height of $h_i$, which can be estimated with Equation (2).

$$h_i \left\{ \ln\left(\frac{h_i}{z_{0\,max}}\right) - 1 \right\} = BkL \tag{2}$$

Here, $B = 1.1$, $k$ is the von Karman constant (= 0.4), $z_{0max}$ is the largest roughness length in the grid box, and $L$, the representative horizontal length scale of a patch, is calculated as $r_i \wedge$, where $r_i$ represents the fraction of each LU type within the patch and $\wedge$ is the patch length. The patch length is identified as the resolution of the grid box (3 km in this study). Additionally, the value of $h_{ei}$ is determined according to the ratio of $h_{ei}/h_i$, for which the prescribed values are 0.05 for a smooth patch and 0.1 for a rough patch [14]. A grid box in which the sum of the areas with $z_{0i}$ values larger than 0.1 m accounts for less than half of the patch area is described as a smooth patch. In contrast, a grid box in which the sum of the areas with $z_{0i}$ values larger than 0.1 m accounts for more than half of the patch area is defined as a rough patch. Finally, the height of the blending layer, $h$ in Equation (1), is set equal to the maximum of all $h_i$ values obtained over the patch.

In the real world, the same LU type may be separated by other LU types within a grid box [12]. This situation would introduce difficulty in defining the representative horizontal length scale of the patch using the method of Goode and Belcher [14]. For simplification, a subtile concept was introduced based on Niu et al. [16]. With the subtile approach, the same LU type within a grid box is aggregated to represent the heterogeneity of the land surface, and $L$ is calculated based on the total proportion and grid spacing.

In this study, we considered only $z_{0eff}$ under neutral conditions, although it has been reported that $z_{0eff}$ is not a constant but a function of stability. Mason [17] noted that buoyancy fluxes tend to reduce $z_{0eff}$ values and fluxes under unstable conditions and increase them under stable conditions. Recently, Zhong et al. [18] explained that although $z_{0eff}$ shows a slight dependence on the atmospheric stability, $z_{0eff}$ can be treated as a constant in the initialization process and can remain unchanged throughout the numerical simulation integration period. This ability is because the variations in the drag coefficients affected by the values of $z_{0eff}$ are less than 2% under all stability conditions. Hence, the $z_{0eff}$ values obtained under neutral conditions were used for the model simulations.

For the meteorological simulations, the Noah land surface model (LSM) [1] was chosen to provide the surface turbulent fluxes and soil moisture for the WRF simulations. In the Noah LSM, $z_0$ values are obtained directly from an LU lookup table based on the dominant LU type, as listed in Table 1 (under the "default" column). Two sets of $z_{0eff}$ values were prepared for this study. The first $z_{0eff}$ dataset was calculated using the $z_{0i}$ values supplied from the dominant LU lookup table (the "default" column in Table 1), while the second $z_{0eff}$ dataset was estimated using the updated $z_{0i}$ values, for which the observed $z_{0i}$ values of three LU types (urban, dry cropland and pasture, and irrigated cropland and pasture) were determined based on observed surface fluxes and wind, TA, and humidity profiles at three sites in Taiwan [13,19]. The method used to calculate $z_{0i}$ was first proposed in Tsuang et al. [19] and further applied for different LU types in Tsai et al. [12,13]. Assuming that the conditions in the atmospheric inertial sublayer (ISL) are steady and horizontally homogeneous, the $z_{0i}$, zero-displacement height, friction velocity, surface SHF, surface LHF, and Monin–Obukhov stability length can all be determined by fitting the integral forms of Businger's equations [20]. Then, profiles of the wind speed, potential temperature, and specific humidity can be used to obtain the maximum correlation between the calculated and observed wind speeds within the height range of the ISL [12,19,21–23].

The updated $z_{0i}$ value of the cropland/grassland mosaic LU type was determined as the mean $z_{0i}$ value between the dry cropland and pasture (0.07 m) LU type and the grassland (0.12 m) LU type. The updated $z_{0i}$ values are listed in Table 1 under the "updated" column. If no value is listed in this column (due to the lack of observational data), the default value from the WRF modeling system [24] was used. It should be noted that summertime LU lookup table values were adopted in this study.

**Table 1.** Default and updated $z_{0i}$ (unit: m) of each LU type.

| LU Type | Default | Updated |
|---|---|---|
| Urban | 0.8 | 2.1 |
| Dry cropland and pasture | 0.15 | 0.07 |
| Irrigated cropland and pasture | 0.10 | 0.03 |
| Cropland/grassland mosaic | 0.14 | 0.095 |
| Cropland/woodland mosaic | 0.2 | |
| Grassland | 0.12 | |
| Shrubland | 0.05 | |
| Mixed shrubland/grassland | 0.06 | |
| Savanna | 0.15 | |
| Deciduous broadleaf forest | 0.5 | |
| Deciduous needleleaf forest | 0.5 | |
| Evergreen broadleaf | 0.5 | |
| Evergreen needleleaf | 0.5 | |
| Mixed forest | 0.5 | |
| Water bodies | 0.0001 | |
| Herbaceous wetland | 0.2 | |
| Barren or sparsely vegetated | 0.01 | |
| Snow or ice | 0.001 | |

Figure 3 shows a comparison among the $z_0$ values at a grid spacing of 3 km. In Figure 3a, the $z_0$ values are obtained from the LU lookup table and are dependent on the dominant LU types (Table 1, "default" column). Figure 3b shows the $z_{0eff}$ values estimated based on the $z_{0i}$ values obtained from the default LU table, while Figure 3c shows the $z_{0eff}$ values estimated based on the updated $z_{0i}$ values. Figure 3a shows the least heterogeneity. A comparison between Figure 3a,b shows a higher degree of spatial variability in Figure 3b, which considers the subgrid-scale variations in the underlying roughness elements. This comparison also shows that the distribution of $z_{0eff}$ is enhanced in most areas of western Taiwan; however, over the major urban areas (e.g., Taipei, Taichung, Tainan, and Kaohsiung), the $z_{0eff}$ values are lower than those obtained from the LU table (Figure 3b,d), which can be explained by Equations (1) and (2).

The $z_{0eff}$ values calculated with the updated $z_{0i}$ (Figure 3c) are especially high in the major urban areas (e.g., Taipei, Taichung, Tainan, and Kaohsiung), where many high-rise buildings are located and where $z_{0eff}$ can even reach 1.8 m. According to Table 1, the updated $z_{0i}$ for the urban LU type is set to 2.1 m, which is much higher than the value from the default LU table (currently 0.8 m). The relatively low $z_{0eff}$ values in parts of western Taiwan also correspond to the lower updated $z_{0i}$ values of cropland-related LU types. Overall, the newly derived $z_{0eff}$ dataset considers the inhomogeneity of the complicated LU pattern and more accurately reflects the real land surface conditions in Taiwan.

Tsai et al. [13] compared estimated $z_{0eff}$ values with observed values from field measurements and concluded that the $z_{0eff}$ results were close to the observed results. In the following, to justify the application of Goode and Belcher's method to high-resolution regional simulations in Taiwan, the derived $z_{0eff}$ data were compared with observed data from Chaoliao Junior High School (refer to Figure 1 for the site location). The observed surface meteorological variables at this site were further used to validate the model output. The details of the observed data are discussed in the next section. The study site is located in a suburban area in southern Taiwan. Within $3 \times 3$ km$^2$ areas, the LU types consist mainly of buildings (22.22%), irrigated cropland and pasture (22.22%), cropland/grassland mosaic (38.9%), grassland (8.33%), and shrubland (8.33%). The observed data were excluded when the friction velocity is below 0.15 m s$^{-1}$ for weak turbulence or when the wind speed was below 1 m s$^{-1}$ based on the sensor detection limits. In addition, data associated with approximately neutral atmospheric conditions, during which the SHF ranged between 0 and 10 Wm$^{-2}$, were selected. After the above selection processes, the method proposed by Martano [25] was employed to determine $z_0$ using the flux, wind speed and TA data. The observed $z_{0i}$ value derived for the Chaoliao site is 0.39 m.

At the Chaoliao site, the dominant LU type is cropland/grassland mosaic, the $z_0$ value from the LU table is 0.14 m, and the estimated $z_{0eff}$ values calculated via Equation (1) using the default and updated LU table values for this grid are 0.34 and 0.53, respectively. Compared with the observed value, the estimated $z_{0eff}$ based on the updated table is too high because the updated $z_{0i}$ for the urban LU type was acquired from a highly compact urban environment. In contrast, the fraction of the urban LU type at the Chaoliao site is only 22.22%. This problem can be fixed if an observed $z_{0i}$ value can be obtained for a low-density urbanized area. Overall, the results reveal that Equation (1) provides reasonable estimates of $z_{0eff}$.

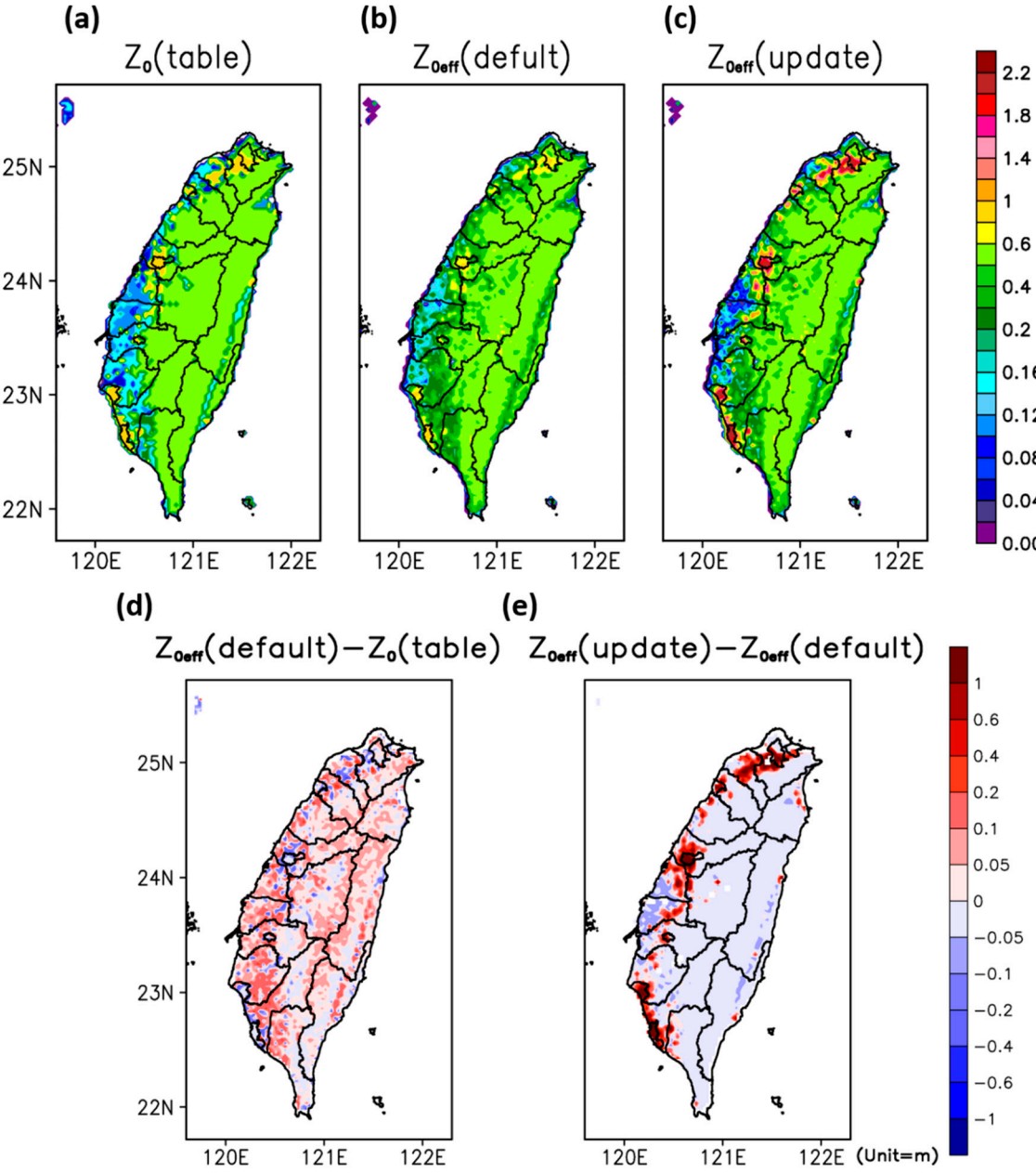

**Figure 3.** (**a**) $z_0$ values from the lookup LU table, (**b**) $z_{0eff}$ values estimated using the default $z_{0i}$, (**c**) $z_{0eff}$ values estimated using the updated $z_{0i}$, (**d**) difference between (**a**) and (**b**), and (**e**) difference between (**b**) and (**c**) (unit: m).

## 3. Descriptions of the Simulation Episode, Observations and Model Configuration

The simulation period ranges from 16 to 30 October, 2011. The October climate in Taiwan still exhibits summer conditions; as a result, summertime LU table values were adopted, as mentioned in the previous section. Throughout the entire study period, the local weather in Taiwan was mostly dominated by a synoptic northeasterly monsoonal flow induced by the Asian continental high-pressure system. Figure 4 shows surface weather maps generated from the National Centers for Environmental Prediction (NCEP) Final Analysis (FNL) data at 08:00 local standard time (LST) on 22–25 October, 2011. On 22 October, a weak continental high-pressure system affected Taiwan, and the prevailing wind in Taiwan was dominated by the northeasterly monsoonal flow. With the eastward movement of the continental anticyclone system from the east coast of China to the East China Sea, the prevailing wind changed from a northerly flow to an easterly flow on 23 October. The high-pressure system moved further away from Taiwan. With the weak influence of the synoptic weather system, the wind speed was relatively weak in western Taiwan on 24 October. On 25 October, another anticyclone system originating from northern China gradually moved towards the east coast of China, putting Taiwan under the influence of a northeasterly flow.

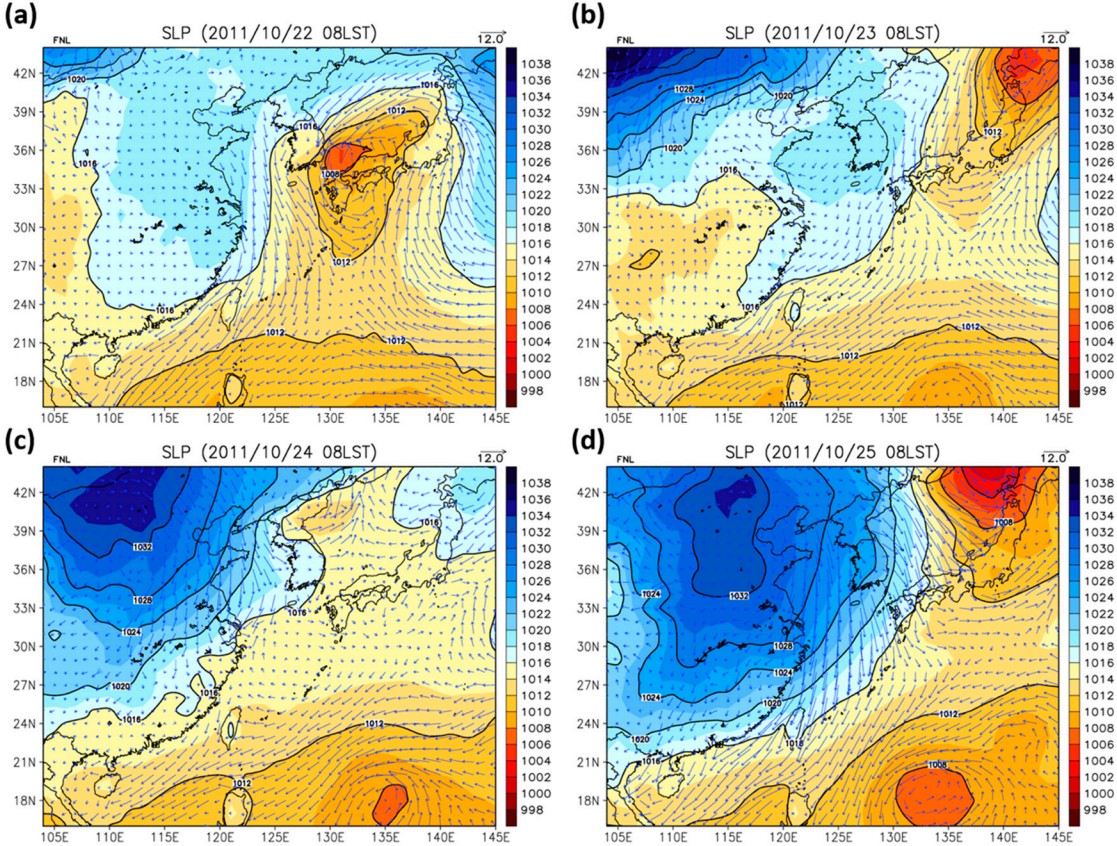

**Figure 4.** Sea level pressure (SLP) (hPa) and surface wind fields (m/s) at 08:00 local standard time (LST) from (**a**) 22, (**b**) 23, (**c**) 24 and (**d**) 25 October, 2011.

During the study period, high ozone ($O_3$) concentrations were observed in southwestern Taiwan. Figure 5 shows the observed $O_3$ concentration and wind fields from surface air quality monitoring stations at 13:00 LST on 24 and 25 October, 2011. As a result of the terrain blocking phenomenon affecting the easterly synoptic wind, western Taiwan exhibited weak wind fields that led to the accumulation of high $O_3$ concentrations on 24 October; consequently, on this date, the $O_3$ problem was the worst throughout the study episode, particularly over southwestern Taiwan. On 25 October,

the enhanced northeasterly monsoonal flow reduced the $O_3$ concentrations over northern Taiwan; however, the $O_3$ concentrations were still high in southwestern Taiwan.

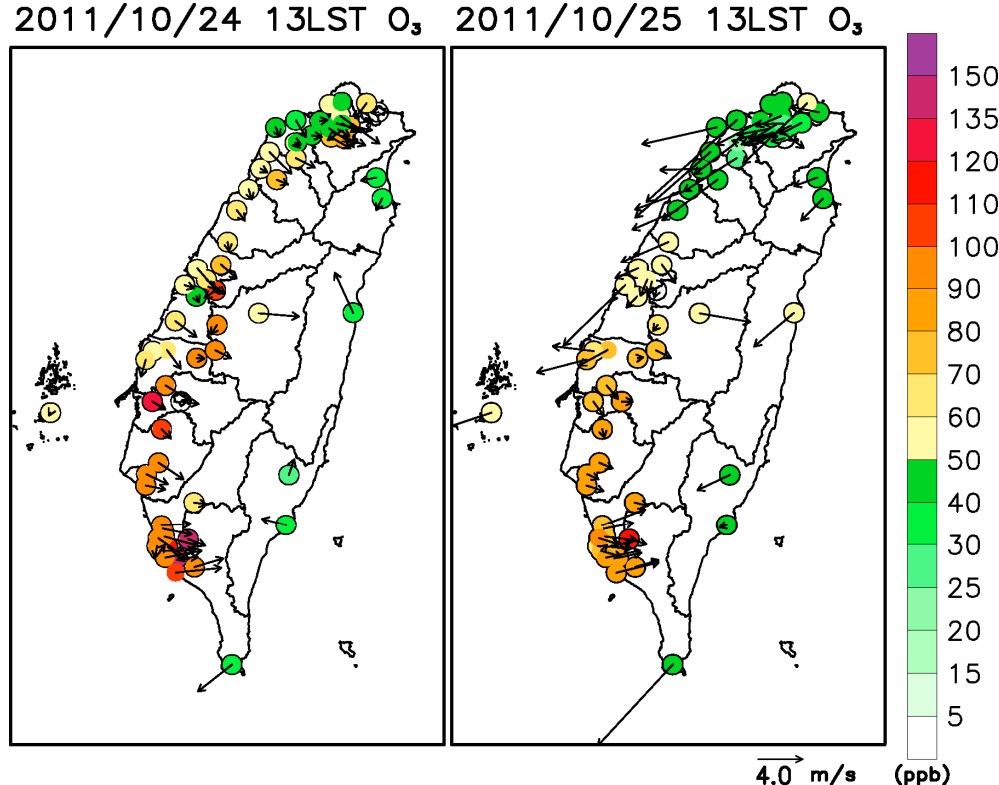

**Figure 5.** Observed $O_3$ concentrations and wind fields from surface air quality monitoring stations at 13:00 LST on 24 and 25 October, 2011.

Southwestern Taiwan is notorious for its variety of heavy industry, including oil refineries, petroleum, iron, and steel manufacturing industries, and power plants. According to Cheng and Hsu [26], southwestern Taiwan frequently encounters serious air pollution problems not only due to emissions but also due to unfavorable meteorological conditions. To investigate the abovementioned high-$O_3$ event and the corresponding meteorological characteristics in the planetary boundary layer (PBL), a field campaign was conducted at Chaoliao Junior High School located in the city of Kaohsiung in southwestern Taiwan. Tethersonde measurements were launched from ground level to a height of approximately 1200 m to measure the boundary layer wind and TA fields at Chaoliao Junior High School. In addition, flux tower measurements were carried out to observe the surface turbulent flux components. An $O_3$ sounding system was also launched (these data were not used in this study). The characteristics of the $O_3$ concentration are not discussed further here but are addressed in another study conducted to investigate the relationship between the meteorological conditions and the air pollution problem in southern Taiwan. In addition to the data collected in Chaoliao, CWB surface weather stations (refer to Figure 1 for the site locations) were used to assess the model performance.

In this study, WRF version 3.4.1 was deployed for the meteorological simulation. Figure 6 shows the model domain setup. The outermost domain covers a large area, and the grid spacing is set at 81 km to reduce the computational time. Next, smaller nested domains were applied with grid spacings successively decreasing down to 27, 9, and 3 km. The vertical resolution was composed of 49 sigma levels, 20 of which were within the lowest 1 km, with the lowest layer at approximately 16 m (full sigma level). The initial and boundary conditions were acquired from the NCEP FNL 6 h reanalysis fields at a one-degree grid spacing. Analysis nudging was applied above the boundary layer

for the wind, TA and water vapor fields through domains 1–3 to utilize a larger-scale forcing to nudge the model simulation towards the reanalysis field.

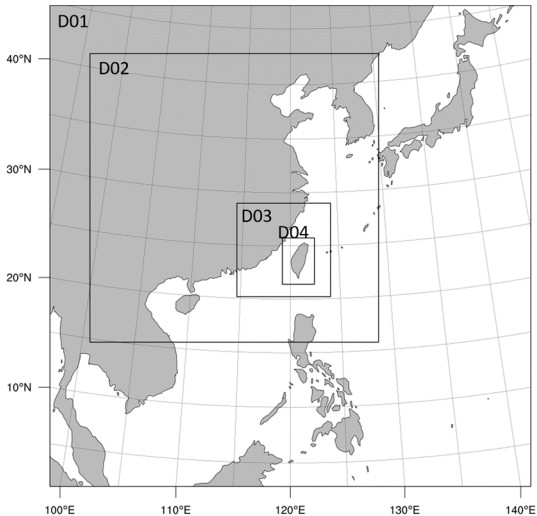

**Figure 6.** Model simulation domains: the outermost domain (**D01**) is with 81 km grid spacing; the nested domains, (**D02**), (**D03**), and (**D04**), are with 27, 9, and 3 km grid spacing, respectively.

The physical options used for the WRF simulations were the Rapid Radiative Transfer Model (RRTM) longwave [27] and simple cloud-interactive shortwave schemes [28] for the radiation process, the Yonsei University PBL scheme [29], and the Kain-Fritsch cumulus parameterization scheme [30,31]. It was assumed that this model is able to resolve organized convection at a fine grid resolution, and thus, the cumulus scheme was not used in the 3 km domain. The Noah LSM [1] was chosen to simulate surface evapotranspiration and runoff processes. The LU data were provided from the 2009 reclassified MODIS LU datasets. The application of these updated MODIS LU data has provided improved results for WRF simulations in Taiwan [32].

Three WRF sensitivity experiments were performed, involving (1) the dominant LU table-based $z_0$ (namely, S1), (2) the $z_{0eff}$ estimated from the default $z_{0i}$ (namely, S2), and (3) the $z_{0eff}$ estimated from the updated $z_{0i}$ (namely, S3). The comparison of the WRF sensitivities between S1 and S2 reveals the effect of considering subgrid variability in the underlying roughness elements, while the comparison between S2 and S3 reveals the effect of utilizing the observed $z_{0i}$ for a few LU categories.

## 4. Simulation Results

### 4.1. Spatial Distributions of the SHF, Temperature, and Wind Fields

This spatial analysis focuses on 24 and 25 October, which were associated with different synoptic weather systems, to demonstrate the impacts of the $z_{0eff}$ on the local meteorological characteristics. A weak synoptic weather forcing and a local land-sea breeze circulation that formed over western Taiwan were associated with 24 October. On the other hand, 25 October was affected by strong northeasterly monsoonal flow. Figure 7 shows the SHF distributions of the first model-level output at 00:00 and 12:00 LST on 24 and 25 October, 2011. The results of the S1 simulation and two difference plots are presented for each time. The S2–S1 plot shows the difference between the S2 and S1 simulations (positive values indicate higher values in S2; a similar scheme is used in all the difference plots). Figures 8 and 9 are similar to Figure 7 but for the TA and wind fields, respectively.

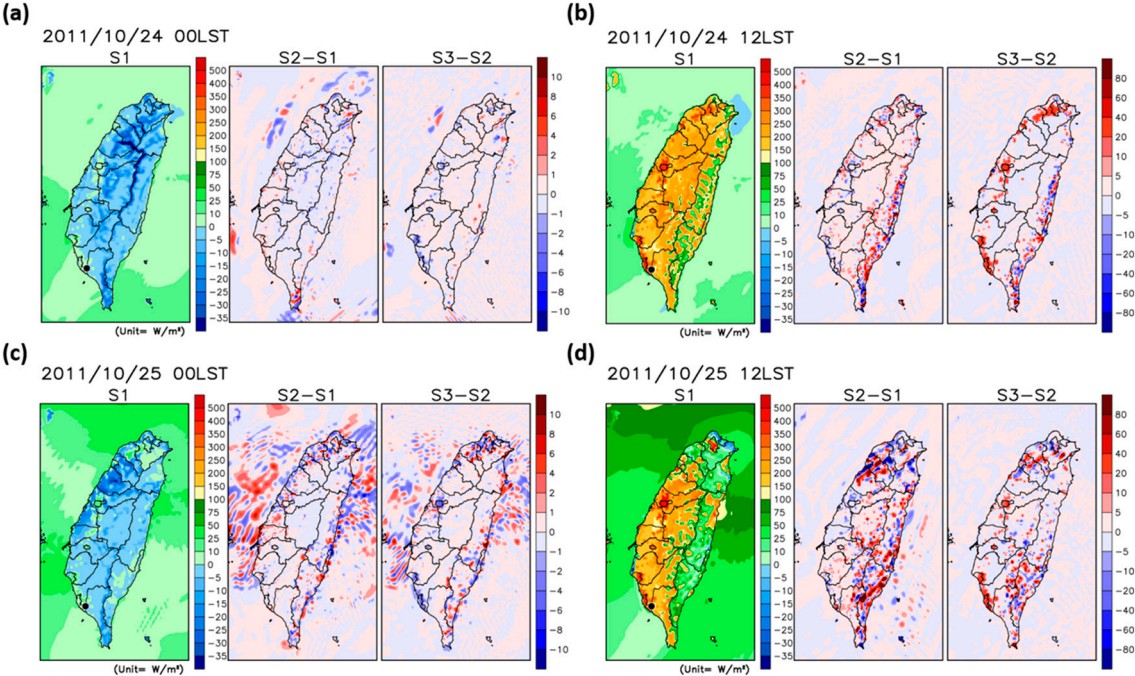

**Figure 7.** Near-surface sensible heat flux (SHF) distribution (units: W/m$^2$) from the S1 simulation, the difference between the S2 and S1 simulations, and the difference between the S3 and S2 simulations. (**a**) and (**b**) is produced at 00:00 and 12:00 LST, respectively on 24 October. (**c**) and (**d**) is produced at 00:00 and 12:00 LST, respectively on 25 October. The circle marks the location of Chaoliao.

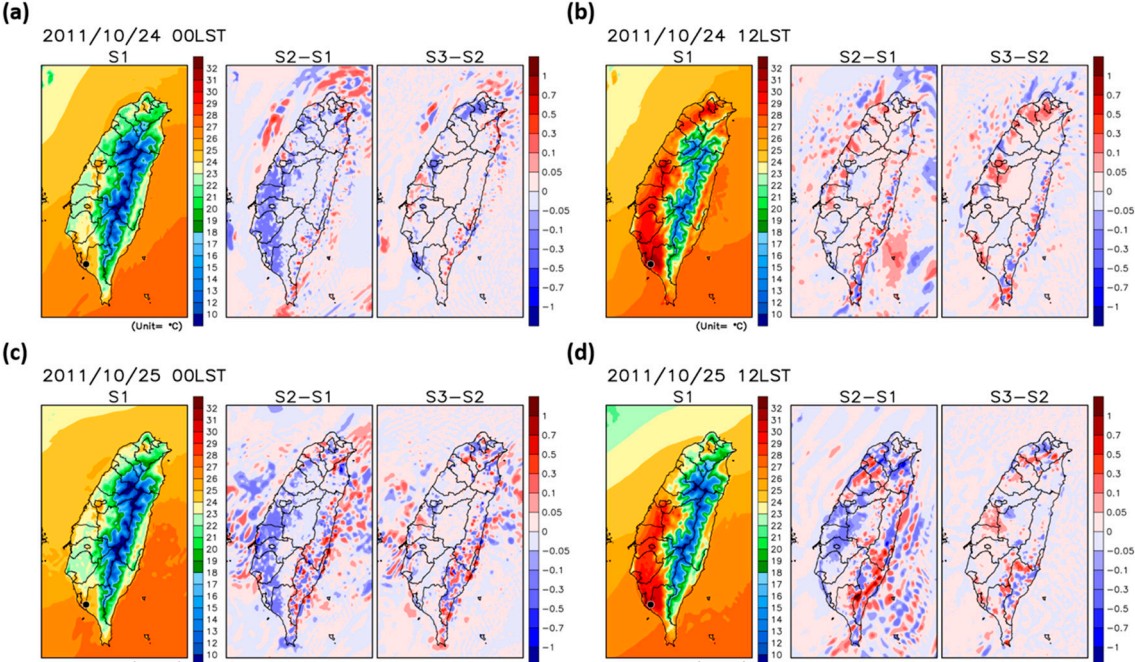

**Figure 8.** Near-surface air temperature (TA) distribution (unit: degrees Celsius) from the S1 simulation, the difference between the S2 and S1 simulations, and the difference between the S3 and S2 simulations. (**a**) and (**b**) is produced at 00:00 and 12:00 LST, respectively on 24 October. (**c**) and (**d**) is produced at 00:00 and 12:00 LST, respectively on 25 October. The circle marks the location of Chaoliao.

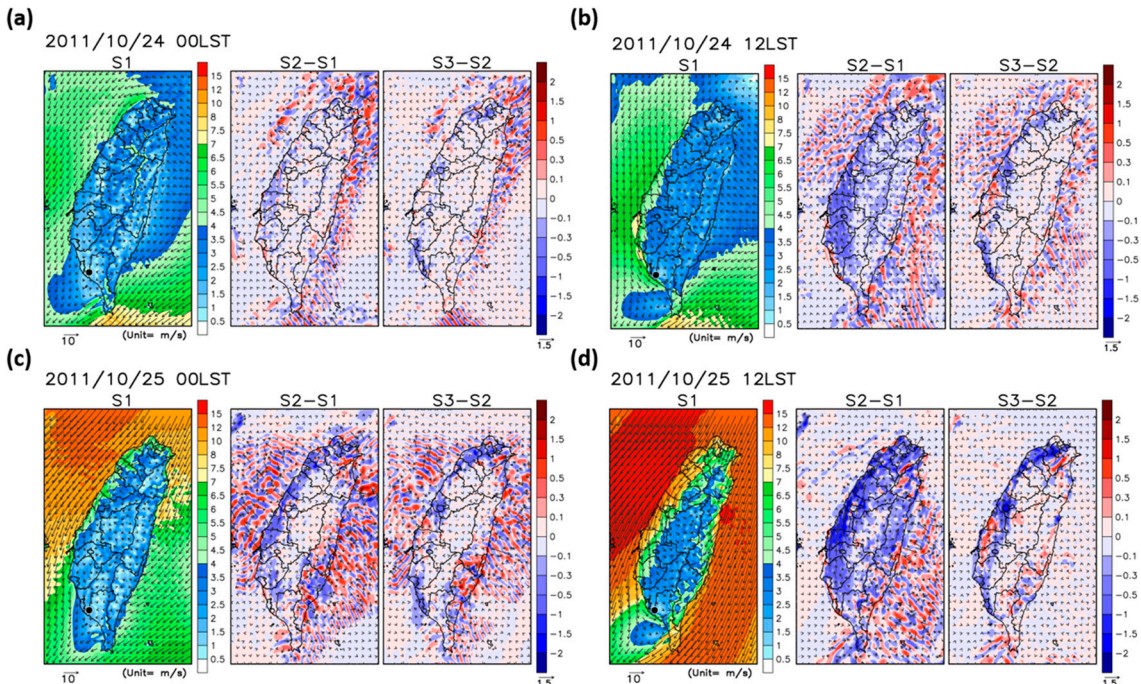

**Figure 9.** Near-surface wind field distribution (units: m s$^{-1}$).from the S1 simulation, the difference between the S2 and S1 simulations, and the difference between the S3 and S2 simulations. (**a**) and (**b**) is produced at 00:00 and 12:00 LST, respectively on 24 October. (**c**) and (**d**) is produced at 00:00 and 12:00 LST, respectively on 25 October. The circle marks the location of Chaoliao.

The direction of the SHF at night is downward with negative values over Taiwan, except for northern Taiwan and some of the central and southern metropolitan areas. The difference between the model simulations on the night of 24 October is small with a magnitude of less than 5 W/m$^2$. During the day, the direction of the SHF transfer is upward. On 24 October, the difference in the SHF between S1 and S2 resembles the difference in $z_0$ between the two models, except in some areas of eastern Taiwan. S3 predicts the highest SHF over urbanized areas due to the enhanced $z_0$. However, on 25 October, due to the enhanced northeasterly synoptic wind flow, the influence of $z_{0\text{eff}}$ on the daytime SHF is less relevant, particularly in northern and eastern Taiwan, possibly due to the complex interaction between the synoptic wind flow and the local surface turbulent fluxes. From the central to the southern part of western Taiwan, where the influence of the synoptic wind flow was weaker, the impact of $z_{0\text{eff}}$ on the simulated SHF is prominent.

On 24 October, between S1 and S2, the nighttime TA is lower in the S2 simulation due to the general increase in $z_0$. Between S2 and S3, the discrepancies are greater over urban areas, where significant increases in $z_0$ appear. Over the major metropolitan areas, the nighttime TA is the lowest in the S3 simulation because $z_0$ is the highest, which enhances the downward transfer of heat flux and reduces the near-surface TA. In contrast to the nighttime distribution, during the day, S3 predicts a higher TA over urban areas than the other two simulations because the upward heat flux is enhanced with the updated $z_{0\text{eff}}$ data. The model comparison on 25 October is similar to that on 24 October; however, the variation in the daytime TA between different simulations does not reveal the apparent influence of $z_{0\text{eff}}$ Overall, the differences in the near-surface TA distributions are not significant among the WRF sensitivity experiments with a magnitude of less than 0.5 degrees Celsius.

For the comparison of the wind field at each time, the northeasterly flow prevailed over the Taiwan Strait (the strait separating the island of Taiwan from Mainland China). Southwestern Taiwan is situated on the leeside of the mountain, where stagnant wind often reduces the dispersion of air pollutants. Stagnant winds were simulated over southwestern Taiwan and its offshore areas on both days (24 and 25 October). According to Hsu and Cheng [33], the frequent occurrence of low wind speed

conditions reduces the dispersion of air pollution, leading to the accumulation of air pollutants near the emission source regions in southwestern Taiwan. As shown in the "S2-S1" plot at 00:00 LST, there is an apparent decrease in the wind speed in the S2 simulation, particularly in western Taiwan. This reduction is more significant during the day than during the night because of the stronger turbulent mixing and exchange processes occurring within the well-mixed daytime boundary layer, which is closely associated with the length scales of the roughness elements. The comparison between the S2 and S3 simulations reveals a further reduction in the wind speed in the latter over urbanized areas. The wind flow was stronger on 25 October than on 24 October; additionally, the reduction in the wind speed due to the enhanced $z_{0eff}$ is more apparent on 25 October. Overall, the S3 simulation using the updated $z_{0eff}$ dataset reflects the lowest surface wind speeds over western Taiwan.

### 4.2. Time Series Comparison

Figure 10 shows a comparison among the time series of the temperature, wind speed and wind direction from 20 to 27 October at the Chaoliao site. The MODIS LU classification identifies this area as being dominated by the cropland/grassland LU type. The $z_0$ value for this LU type used in each simulation was 0.14 for S1, 0.34 for S2, and 0.53 for S3. The difference in temperature among the three simulations was very small, but there was an overestimation of the wind speed in the S1 simulation. With the use of $z_{0eff}$, both S2 and S3 yielded lower wind speeds that agree better with the observed data. The observed winds revealed a weak northwesterly flow throughout the nighttime and morning hours, and all model runs were able to capture the observed pattern. There was an apparent turning of the wind towards a southwesterly onshore direction in the afternoon throughout the study period in the observed datasets. All the simulations behaved similarly and captured the turn in the wind direction well, except during the afternoon hours on 20, 22, and 27 October. Moreover, the daytime sea breeze flow was better captured than the nighttime wind.

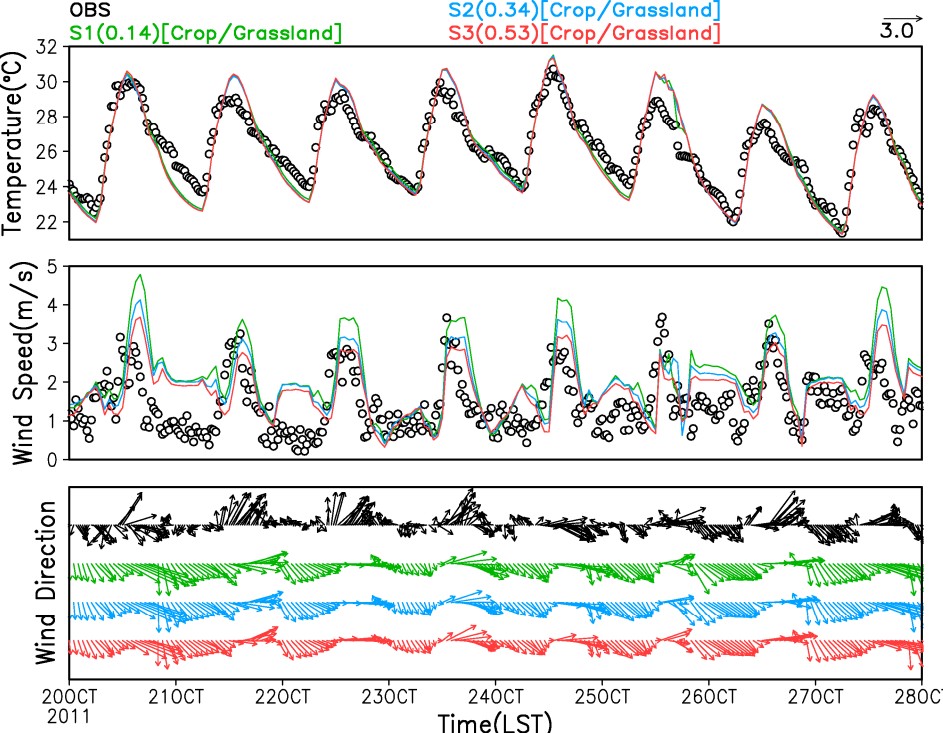

**Figure 10.** Time series comparison of the temperature (upper panel), wind speed (middle panel), and wind direction (bottom panel) among the observations (black) and the S1 (**green**), S2 (**blue**), and S3 (**red**) sensitivity simulations.

Figure 11 is similar to Figure 10 but shows the SHF, LHF, and frictional velocity (UST). The simulations using $z_{0eff}$ (S2 and S3) exhibited slightly higher surface heat flux exchange processes, with differences reaching 5 W/m$^2$ during the day and very small differences during the night. S1 showed a significant underestimation of the UST, while the S2 and S3 simulations yielded higher UST values that agreed better with the observed data. All the simulations underestimated the nighttime momentum flux.

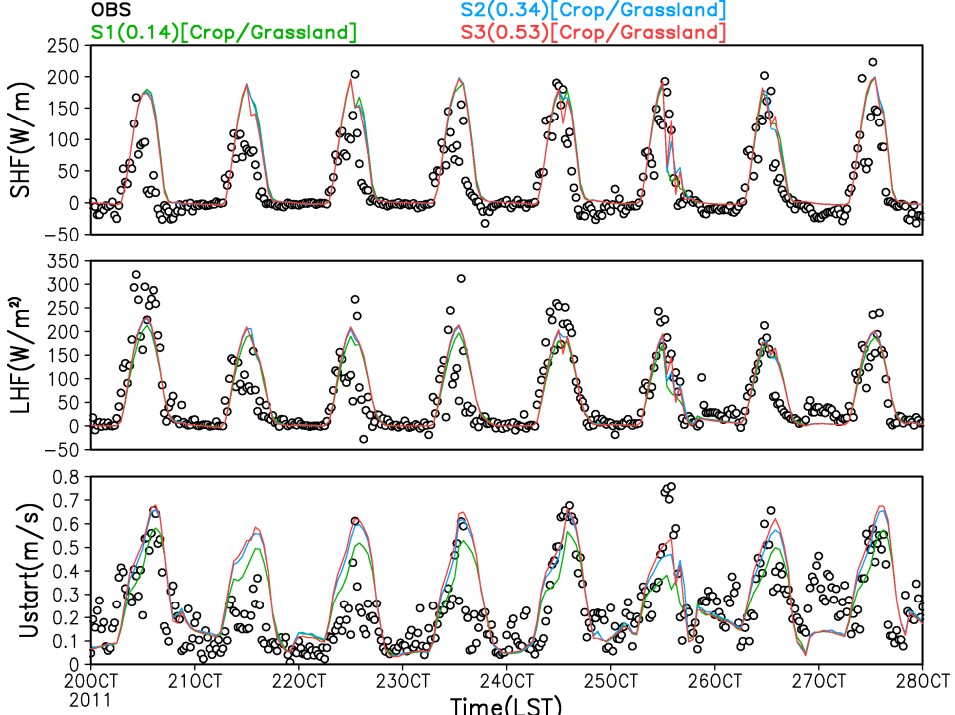

**Figure 11.** Similar to Figure 9 but for the SHF (**upper panel**), latent heat flux (LHF) (**middle panel**) and frictional velocity (UST) (**bottom panel**).

The above analysis shows that the implementation of $z_{0eff}$ successfully reduced the overestimation of the wind speed and underestimation of the surface momentum flux. In reality, Chaoliao Junior High School is surrounded by cropland and several building structures. The updated $z_{0eff}$, which considers heterogeneous land surface structures, represents the roughness elements in the study area better than the data provided from the lookup table. With more accurate $z_0$ data, the S3 simulation showed good performance in terms of the wind field and surface momentum flux predictions.

*4.3. Comparison with the Tethersonde System*

Figure 12 shows the potential temperature profiles from the tethersonde system observations made on 24 and 25 October at the Chaoliao site. The evolution of the temperature profiles, along with development of the PBL, is clearly observed on both days. At 06:00 LST, the underlying surface was colder than the air; a stable boundary layer (SBL) formed near the surface layer, and the potential temperature increased with height. At 08:00 LST, the surface warmed due to heating from solar radiation, and the mixing layer began to grow in depth. At 12:00 LST, the SBL completely disappeared, and a well-mixed boundary layer formed; in addition, the mixing layer continued to grow in depth. After sunset, the surface cooled again, and the 18:00 LST profile shows that a stable layer formed near the surface layer. The 22:00 LST profile shows the nighttime SBL that formed near the surface; moreover, above the SBL, a neutral residual layer appeared where thermals ceased to form and exhibited the characteristics of the previous daytime mixing layer.

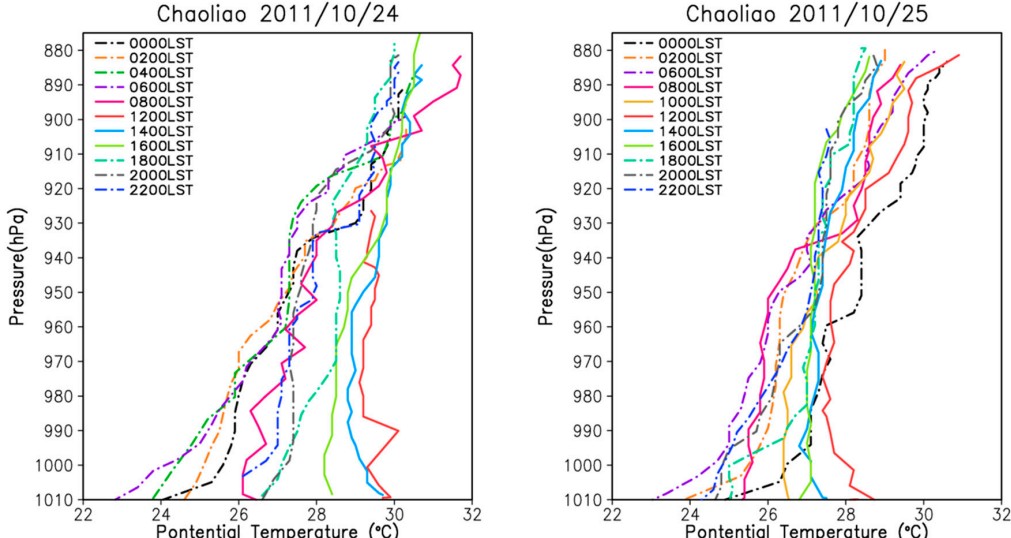

**Figure 12.** Observed potential temperature profiles at the Chaoliao site on 24 and 25 October, 2011.

Figure 13 shows a comparison among the vertical potential temperature profiles of the observations and the simulation outputs. An apparent temperature inversion layer formed during the nighttime; however, the strength of the simulated SBL was weaker than the strength of the observed SBL. During the day, all the simulations behaved quite similarly: all the models simulated a slightly cooler daytime structure on 24 October and an excessively strong daytime mixing layer on 25 October. For the comparison among the wind fields, all the simulations predicted an excessively strong northeasterly monsoonal flow and failed to capture the wind variations during the night (figure not shown). The comparison among the vertical profiles from the different simulations does not reveal apparent discrepancies, indicating that the impact of $z_0$ is limited mostly near the surface layer.

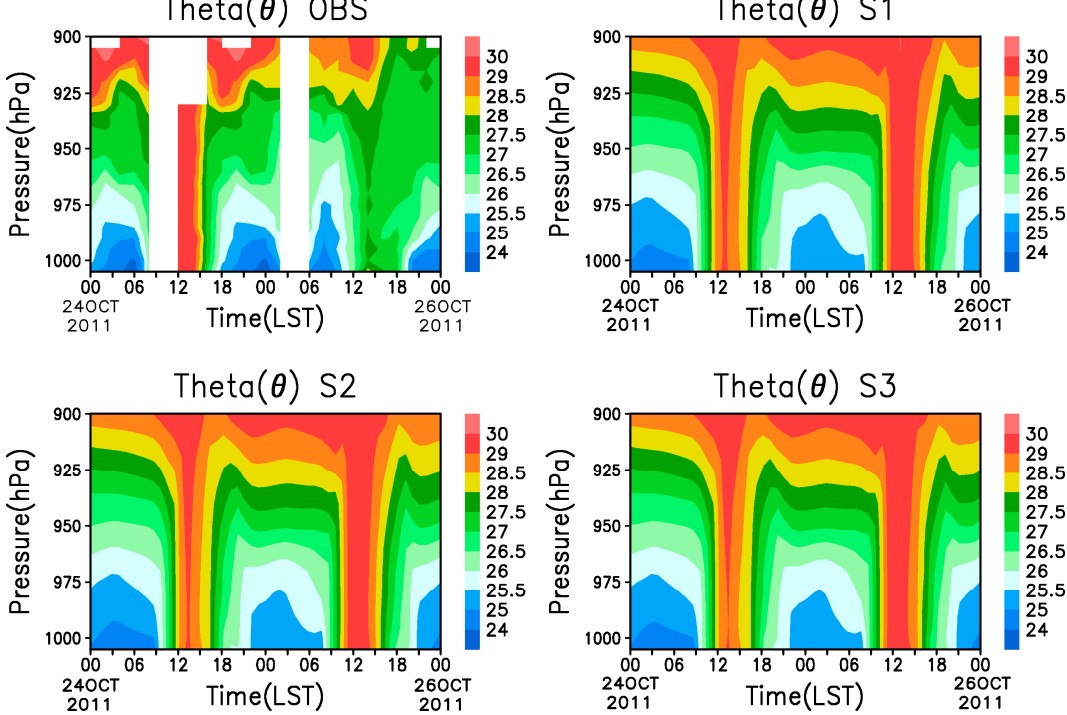

**Figure 13.** Comparison among the vertical potential temperature profiles (unit: degrees Celsius) based on the observations and the S1, S2, and S3 simulations on 24 and 25 October, 2011 at the Chaoliao site.

### 4.4. Statistical Analysis

Figure 14 shows a comparison between the mean bias (MB) and root mean square error (RMSE) for the TA and wind speed for all the CWB surface stations (refer to Figure 1 for the site locations) in Taiwan from 17 to 28 October, 2011. The comparative analysis is performed according to the number of different LU types in the grids closest to the CWB sites; this number is identified along the *x*-axis of the figure. For example, if the number of LU types is equal to one, the analysis for the CWB site has only one LU composition; in other words, the analysis of the site reflects the least land surface heterogeneity. If the number of LU types is equal to seven, the analysis of the CWB site has seven different LU compositions; in other words, the land surface at this site is quite heterogeneous. The statistical scores of the TA among the three simulations are comparable to each other, with slightly lower bias and RMSE values for the S3 simulation; the performance of the S2 simulation is also slightly better than that of the S1 simulation. In the wind speed comparison, S2 and S3 both show improvements for the sites composed of more than 5 LU types. The S3 simulation shows the best performance among the three simulations; the improvement is apparent at sites that exhibit high land surface heterogeneity. Overall, the results of the statistical analysis demonstrate that the implementation of the updated $z_{0\text{eff}}$ successfully reduced the wind speed overestimation.

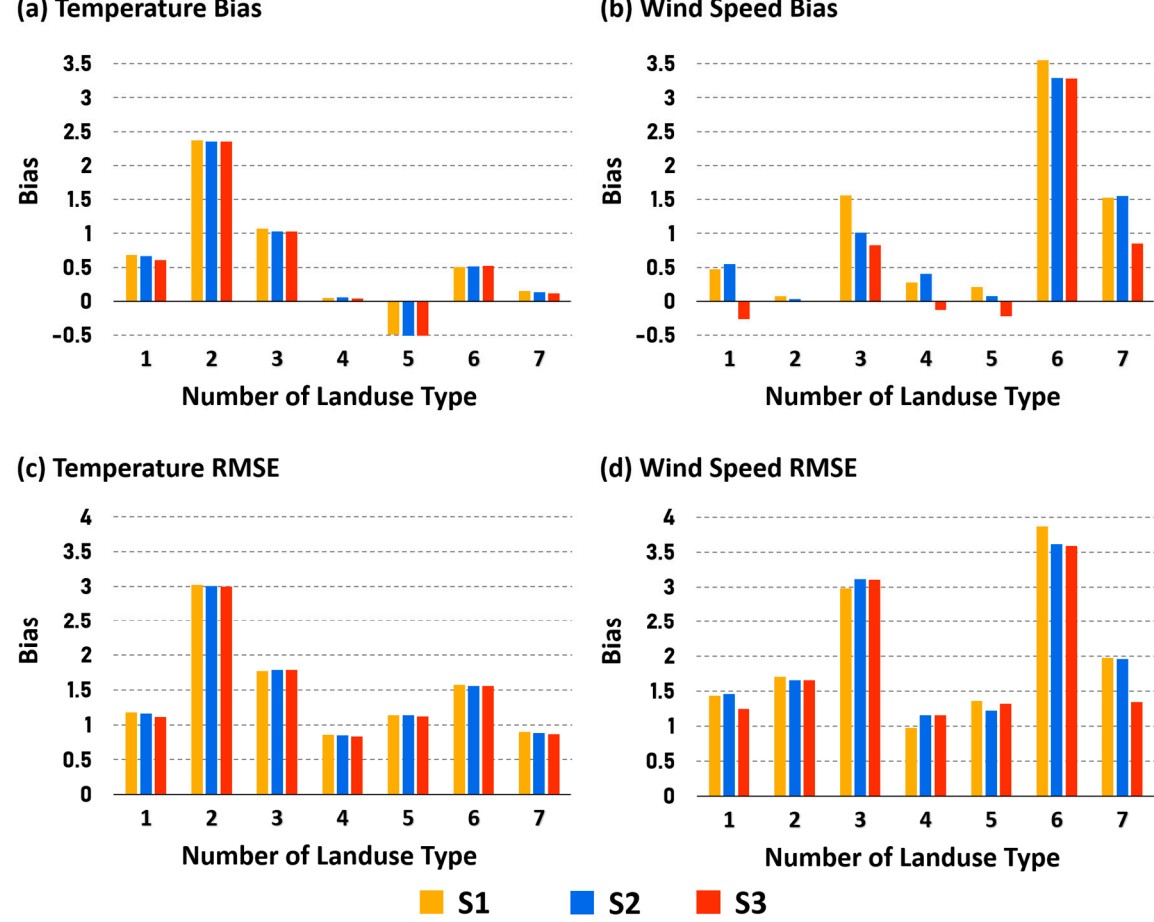

**Figure 14.** Comparison between the mean bias (MB) and root mean square error (RMSE) for (**a**), (**c**) TA (unit: degrees Celsius) and (**b**), (**d**) wind speed (unit: m/s) for the CWB surface stations in Taiwan. The yellow bars are for the S1 simulation, the blue bars are for the S2 simulation, and the red bars are for the S3 simulation.

Moreover, the Pearson correlation coefficient (R), MB, normalized mean error (NME), and RMSE were estimated to provide a comprehensive assessment of the model performance. Table 2 presents the

statistical scores of the simulated TA and wind speed. The simulated TA showed good agreement with the observations. In contrast, all the simulations tended to overestimate the surface wind speed. In general, the S3 simulation showed the best statistical results. Both the MB and the RMSE were the lowest in the S3 simulation.

**Table 2.** Statistical scores of the simulated TA and wind speed (WS) from 17 to 28 October, 2011.

|  | Average | R | MB | NME | RMSE |
|---|---|---|---|---|---|
| **TA (°C)** |  |  |  |  |  |
| OBS | 22.32 | - | - | - | - |
| S1 | 23.07 | 0.823 | 0.749 | 0.961 | 1.09 |
| S2 | 23.05 | 0.822 | 0.735 | 0.961 | 1.088 |
| S3 | 23.03 | 0.824 | 0.71 | 0.95 | 1.077 |
| **WS (m/s)** |  |  |  |  |  |
| OBS | 3.465 | - | - | - | - |
| S1 | 4.63 | 0.580 | 1.167 | 1.573 | 1.707 |
| S2 | 4.5 | 0.579 | 1.039 | 1.499 | 1.641 |
| S3 | 4.19 | 0.579 | 0.724 | 1.364 | 1.507 |

*4.5. Discussion*

Although only 15 days were investigated to study the impact of $z_{0\text{eff}}$, we believe that the simulation results adequately provided sufficient evidence to support the outcomes of this study. During the 15 days study episode, different weather patterns affected Taiwan, including northeasterly monsoonal flow, continental high-pressure system reflux, weak synoptic weather, and local land-sea breeze circulation. In other words, multiscale weather systems occurred in Taiwan, excluding the summer weather pattern. The summer case is of less concern because this study was designed to study the air pollution problem, which occurs mainly from autumn to the following spring.

Moreover, most analyses were conducted for two days, 24 and 25 October, which were associated with completely different synoptic weather systems. A weak synoptic weather system and a prominent local land-air interaction were associated with 24 October; however, 25 October was dominated by strong northeasterly monsoonal flow, and the influences of $z_{0\text{eff}}$ on the surface turbulent fluxes, surface temperature, and wind fields are less relevant.

Finally, $z_{0\text{eff}}$ should be applied in future meteorological model applications in Taiwan to reduce the bias of the wind fields and enhance the modeling capability.

## 5. Conclusions and Future Work

The surface heterogeneity cannot be described properly if the land surface parameters supplied from the lookup table are dependent on the dominant LU type. To account for surface inhomogeneity in mesoscale meteorological simulations, $z_{0\text{eff}}$ was estimated by considering the subgrid variations in the underlying roughness elements based on the LU types within the modeling grid box. However, the USGS 25-category LU dataset is outdated and therefore cannot correctly represent the current land cover distributions in Taiwan. Hence, an updated LU dataset was reclassified using 2009 MODIS satellite products to estimate $z_{0\text{eff}}$.

Comparisons among three simulations show that the enhanced heat flux exchange processes in the surface layer increase the daytime TA and decrease the nighttime TA in the S2 and S3 simulations, particularly over urbanized areas. However, the effect of considering the subgrid heterogeneity on the roughness elements is less significant for thermal fields than for wind fields.

The overestimation of the surface wind speed is improved in the S2 and S3 simulations due to the specification of a higher $z_0$, which enhances the surface drag and momentum fluxes and slows the surface wind flow. The comparison of the results acquired at the Chaoliao site demonstrates the importance of accounting for the subgrid variability in the underlying roughness elements within the

modeling grids. The inclusion of $z_{0\text{eff}}$ improves the wind speed prediction but does not significantly improve the temperature prediction.

The results of this study highlight the importance of utilizing $z_{0\text{eff}}$, which considers the surface heterogeneity and implicitly includes the subgrid variability in the underlying LU types; as a consequence, numerical weather predictions can be improved. These improvements are more prominent in areas where the land surface is highly heterogeneous. Model evaluations at the Chaoliao site and the CWB surface stations further emphasize the need for updated and correct land surface parameters to accurately conduct meteorological modeling. Furthermore, several studies conducted in Taiwan have demonstrated that the WRF meteorological model tends to overestimate the wind speed, which can degrade the air quality model performance [34,35]. Accordingly, the implementation of $z_{0\text{eff}}$ is expected to reduce the surface wind flow and enhance the air quality modeling capability.

**Author Contributions:** Conceptualization, F.-Y.C.; formal analysis, C.-F.L. and Y.-T.W.; investigation, F.-Y.C., C.-F.L. and Y.-T.W.; methodology, F.-Y.C., J.-L.T., and B.-J.T.; supervision, F.-Y.C.; validation, B.-J.T. and C.-H.L.; writing—original draft, F.-Y.C.; writing—review and editing, F.-Y.C. and B.-J.T.

**Funding:** This study was part of the research project "Updated Land Use Data for Taiwan's Meteorological Simulation" supported by the National Science Council, Taiwan, under grant number NSC-98-2111-M-008-024-MY2.

**Acknowledgments:** We thank the Central Weather Bureau and Environmental Protection Agency in Taiwan for providing the surface station datasets.

**Conflicts of Interest:** The authors declare no conflicts of interest.

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
