# Peer review of "Impact of Effective Roughness Length on Mesoscale Meteorological Simulations over Heterogeneous Land Surfaces in Taiwan"

_atmosphere, doi:10.3390/atmos10120805_

Round 1

Reviewer 1 Report

See attached file

Reviewer 2 Report

Reviewer Comments of atmosphere-647712: “Impact of Effective Roughness Length on Mesoscale Meteorological Simulations Over Heterogeneous Land Surfaces in Taiwan” by Fang-Yi Cheng et al. 2019

This paper investigates the roughness length impacts on meteorological simulations in Taiwan. The updated aerodynamic roughness lengths derived from observations are applied in WRF model simulations. Improved model performance in simulating wind speed is achieved based on comparison with observations. I think the manuscript is well written and structured. The topic is related to the land-atmosphere exchange, which is of interest to meteorology and air quality research fields. However, I think there are a couple of places that need clarification before being published. For example, how the individual roughness lengths from different land-use types are estimated from observations. I recommend a major revision for this manuscript. Detailed comments are listed below.

Major Comments:

Calculation of z0i: z0i is one important parameter to calculate z0eff. The authors briefly state that z0i is ‘based on observed surface fluxes and wind, temperature and humidity profiles at three sites in Taiwan’ (Ln 164-166). Considering the importance of this parameter, I think it deserves more detailed descriptions (equations) as to how it is calculated in Section 2. Grid resolution: The WRF domain resolutions used in this paper are 81, 27, 9, and 3km. This seems larger than usual grid resolutions used in air quality (12km CONUS) and meteorology (even smaller than 1km) simulation. Could the authors add justifications for choosing these resolutions? Do authors expect different results regarding to higher grid resolutions, since it will determine the number of different land-use types assigned to each model grid? Comparison with observations: Based on Figure 1 and Figure 14, there are several other observational sites. Could the authors relate the sites in Figure 14 to the spatial locations shown in Figure 1? Maybe adding site codes. It helps for understanding the spatial variations of model performance. Corresponding discussions for different model performances at different sites could be interesting. Why Chaoliao site is chosen for detailed analysis in Section 5.2? Author may consider adding scatter dots from observations in Figure 6-8 to show model spatial variability.

Minor Comments:

Ln 14: Write out WRF. Figure 1: Consider adding the site codes, since they are further discussed in Figure 14. What are the topography heights on the parts of the map that do not have contour lines? Eq1: The equation is not displayed properly. Ln 209: Could the authors add the detailed calculation method for z0i? Figure 4: This figure needs a higher resolution. Ln 257: Please add one extra sentence for why Kain-Fritsch cumulus scheme is not used in the 3-km domain. Figure 6-8: Please consider adding observational data points on the spatial plots. Ln 302: watts/m2-----W/m2. Same in Ln 309. Section 6: The authors mentioned that they are going to use the improved meteorological fields to drive air quality modeling. Could the authors add one or two sentences discussing the implications of this work for air quality modeling?

Round 2

Reviewer 1 Report

I feel satisfied from the answers of the authors. They have included my suggestions and made efforts to improve the paper.

The only question that arises to me is why the authors do not include their extensive and convincing answer of the cover letter at page 4. They explain very well the impact of their work even if based on a limited observational period. I think they could add a subsection like "Discussion of the simulations method and future improvements" (or something similar) in the simulations results, because it is very interesting for me and I think it could be also very interesting for other readers.

Author Response

We have added subsection “Discussions” in Section 4 to discuss the outcomes of this study.

The manuscript also went through a 2nd proof reading service (by AJE) to improve the understanding of this study.

Please refer to the revised version.

Thank you for your comment.

Reviewer 2 Report

My comments were addressed satisfactorily.

Author Response

Thank you for your comment.

The manuscript also went through a 2nd proof reading service (by AJE) to improve the understanding of this study.